# Effects of Paclitaxel on Plasma Membrane Microviscosity and Lipid Composition in Cancer Cells

**DOI:** 10.3390/ijms241512186

**Published:** 2023-07-29

**Authors:** Liubov Shimolina, Alexander Gulin, Alexandra Khlynova, Nadezhda Ignatova, Irina Druzhkova, Margarita Gubina, Elena Zagaynova, Marina K. Kuimova, Marina Shirmanova

**Affiliations:** 1Institute of Experimental Oncology and Biomedical Technologies, Privolzhsky Research Medical University, Minin and Pozharsky Square, 10/1, 603005 Nizhny Novgorod, Russia; shimolina_l@pimunn.net (L.S.); niibmt@pimunn.net (A.K.); ignatova_n@pimunn.net (N.I.); druzhkova_i@pimunn.net (I.D.); 2N.N. Semenov Federal Research Center for Chemical Physics, Russian Academy of Sciences, Kosygin st. 4, 119991 Moscow, Russia; aleksandr.gulin@phystech.edu (A.G.); gubina.mv@phystech.edu (M.G.); 3Lopukhin Federal Research and Clinical Center of Physical-Chemical Medicine, Federal Medical Biological Agency, Malaya Pirogovskaya, 1a, 119435 Moscow, Russia; zagajnova_e@pimunn.net; 4Department of Chemistry, Imperial College London (White City Campus), London W12 0BZ, UK; m.kuimova@imperial.ac.uk

**Keywords:** FLIM microscopy, molecular rotor, microviscosity, plasma membrane, mass spectrometry ToF-SIMS, cancer cells, paclitaxel

## Abstract

The cell membrane is an important regulator for the cytotoxicity of chemotherapeutic agents. However, the biochemical and biophysical effects that occur in the membrane under the action of chemotherapy drugs are not fully described. In the present study, changes in the microviscosity of membranes of living HeLa–Kyoto tumor cells were studied during chemotherapy with paclitaxel, a widely used antimicrotubule agent. To visualize the microviscosity of the membranes, fluorescence lifetime imaging microscopy (FLIM) with a BODIPY 2 fluorescent molecular rotor was used. The lipid profile of the membranes was assessed using time-of-flight secondary ion mass spectrometry ToF-SIMS. A significant, steady-state decrease in the microviscosity of membranes, both in cell monolayers and in tumor spheroids, was revealed after the treatment. Mass spectrometry showed an increase in the unsaturated fatty acid content in treated cell membranes, which may explain, at least partially, their low microviscosity. These results indicate the involvement of membrane microviscosity in the response of tumor cells to paclitaxel treatment.

## 1. Introduction

The plasma membrane is a dynamic, complexly organized cell structure that separates the internal contents of the cell from the external environment. In cancer cells, the membrane is the primary barrier for chemotherapeutic drugs, so its biophysical properties and biochemical composition are important factors that determine efficient drug uptake [1]. On the other hand, chemotherapeutic drugs themselves can alter the microviscosity of the plasma membrane through direct interaction with the lipid bilayer or indirectly through lipid peroxidation [2,3,4]. In addition, fluctuations in membrane microviscosity and changes in lipid profile accompany the response of cancer cells to cytotoxic chemotherapy, including the development of cell death and proliferation arrest. Meanwhile, the effects of chemotherapeutic agents on plasma membranes remain poorly investigated.

Taxol, or paclitaxel, is a chemotherapeutic agent of a plant alkaloids class, which has proven antineoplastic activity against a variety of cancers, including breast, brain, head and neck, lung, colon, cervical, and ovarian tumors [5]. Its major mechanism of action is the stabilization of microtubules via polymerization of tubulin and preventing depolymerization, leading to mitotic arrest and cell death. However, biophysical and biochemical processes that occur before and after the binding of paclitaxel to microtubules are not well understood. Recent studies suggest that the plasma membrane is also the target of taxane drugs. Both membrane composition and fluidity can influence penetration into cells and, consequently, the biological action of paclitaxel. This paclitaxel–lipid interplay is largely determined by phospholipid properties. The higher degree of acyl chain unsaturation, smaller headgroup size, and longer acyl chain length increase paclitaxel incorporation into the lipid membrane [6]. Paclitaxel, due to its lipophilic nature, actively interacts with the cell membrane, and this interaction may contribute to the drug retention and development of chemoresistance [7]. At the same time, taxanes, including paclitaxel, can induce serious physicochemical alterations in membranes, specifically changes in membrane fluidity, conformation of receptors and enzymes, lipid packing density, etc. It is known that paclitaxel causes membrane liquefaction and, conversely, rehydration in a dose-dependent manner [6]. In general, data on the effect of paclitaxel on membrane microviscosity and lipids are scarce and contradictory, and these studies were mainly performed on model membranes.

The aim of the present work was to study microviscosity and lipid changes in the plasma membranes of tumor cells during chemotherapy with paclitaxel.

Membrane microviscosity of live cells in cell monolayers and multicellular spheroids was measured using a two-photon fluorescence lifetime imaging microscopy (FLIM) with the fluorescent viscosity-sensitive dye BODIPY 2. FLIM of BODIPY-based fluorescent molecular rotors is an established technique for measuring viscosity at the microscopic level that allows both quantitative imaging of viscosity with high spatial resolution and dynamic measurements of viscosity in a living cell with high temporal resolution [8,9]. The fluorescence characteristics of molecular rotors, such as fluorescence intensity and lifetime, are strictly dependent on the viscosity of their immediate microenvironment. Of them, the lifetime-based readout seems more appropriate since it does not depend on the concentration of the rotor and its photobleaching, the intensity of the exciting light, and the configuration of the microscope. Thus, the fluorescence lifetime of a rotor can be directly converted to the viscosity of its medium using pre-recorded calibration curves. The methodologies for imaging of membrane microviscosity in cultured cancer cells, tumor spheroids, and animal tumors in vivo have been previously developed by our group and applied to follow the changes over the course of therapy with platinum drugs and 5-fluorouracil [10,11,12].

In parallel, the membrane lipid composition was identified using time-of-flight secondary ion mass spectrometry (ToF-SIMS) with a focus on phosphatidylcholine, sphingomyelin, cholesterol, and unsaturated fatty acids content, the key components responsible for the regulation of viscosity. ToF-SIMS enables the detection and visualization of organic compounds in cells with submicron spatial resolution. High chemical specificity and sensitivity make ToF-SIMS a valuable tool for the analysis of lipids in the cell plasma membrane. Our previous studies have revealed that changes in lipid profile underlay viscosity changes in cell membranes induced by chemotherapeutic agents [10,11,12].

## 2. Results

### 2.1. Membrane Microviscosity in Monolayer Cells Treated with Paclitaxel

In the first stage, we performed an analysis of changes in the microviscosity of the plasma membranes of tumor cells in a monolayer exposed to three different doses of paclitaxel, 1.6, 3.2, and 6.4 nM (Figure 1). After staining HeLa–Kyoto cells with the BODIPY 2 fluorescent rotor, a pronounced fluorescence of the plasma membranes was observed. The fluorescence decay curve of the rotor in the membrane fitted to a monoexponential model (χ^2^ 0.8–1.2), which is typical for BODIPY 2.

In control cells, the fluorescence lifetime of BODIPY 2 was 2.39 ± 0.05 ns, which, according to the calibration curve, corresponded to a viscosity value of 302 ± 17 cP. After adding paclitaxel, the membrane microviscosity did not change for at least 3 h of incubation. At 6 h, there was a tendency for the membrane microviscosity to decrease, especially combined with the largest dose of the drug (281 ± 19 cP). After 24 h, membrane microviscosity became significantly lower than the control value for all doses used (*p* = 0.00011), with slightly more pronounced changes for the highest dose, 260 ± 15 cP. Typical FLIM images and viscosity quantification at all doses of paclitaxel used are presented in Figure 1.

To characterize the cellular response to paclitaxel, cell viability and morphology were assessed (Figure 1C). After 24 h of exposure, the treated cells showed multiple morphological changes, such as rounded shapes and multinucleation. The number of dead cells increased with an increase in drug dose. At a dose of 1.6 nM (IC50/2), the number of dead cells was 16%; at 3.2 nM (IC50), it increased to 19%; at the highest dose, 6.4 nM (IC50×2), it was about 24% compared with 2% in control (Figure 1D).

To check whether the effects of paclitaxel on the cell membrane are persistent, we additionally measured the microviscosity 48 h after the removal of the drug from the culture medium and found that the microviscosity was retained at the low level of 271 ± 23 cP in the absence of the drug (Appendix A).

To verify that the reduced microviscosity of plasma membranes is a result to the cellular response of the drug, we performed experiments on a HeLa–Kyoto subline adapted to paclitaxel. According to our findings, the morphology of cells adapted to paclitaxel did not differ from control cells that were not adapted to the drug (Figure 2). However, the IC50 value increased from 3.2 ± 0.27 nM to 5.96 ± 0.36 nM, which indicates a decrease in sensitivity to the drug by ~1.86 times. At the same time, we found that the microviscosity of the membranes of cells adapted to paclitaxel was only slightly lower than that of the control cells—273 ± 26 cP vs. 299 ± 30 cP (differences from the control are not statistically significant).

### 2.2. Imaging of Membrane Microviscosity in Tumor Spheroids upon Paclitaxel Treatment

Then, we investigated the changes paclitaxel induced in the membrane microviscosity of 3D tumor spheroids, a more complexly organized in vitro tumor model, which recapitulates intercellular interactions and heterogeneity of a tumor more fully than monolayer cells. To evaluate the response of the tumor spheroids to paclitaxel treatment, the spheroid morphology and viability of their cells were analyzed during 48 h exposure (Figure 3). During the natural growth phase from 5 to 7 days, an increase in size from ~330 µm to ~450 µm in diameter was observed. The control spheroids were represented by dense cell complexes with large nuclei and a loosely packed layer of outer cells (Figure 3A). At the same time, single dead cells were found, and their relative proportion increased from 2% to 6% as spheroids grew (Figure 3E). Although cells at the periphery of the spheroid proliferate more actively, the microviscosity of cell membranes did not differ between the outer and inner layers and equaled 297 ± 28 cP (Figure 3C), which corresponds well to data obtained for the cell monolayer.

Treatment with paclitaxel resulted in an increasing number of dead cells and structural disorganization of the spheroids with a loss of integrity, which could be detected after 24- and 48-h of incubation with paclitaxel (Figure 3B). The proportion of dead cells increased from 3% to 51% (Figure 3F). Paclitaxel caused a significant decrease in microviscosity after 24 h and 48 h to 233 ± 19 cP and 237 ± 18 cP, respectively (Figure 3D).

### 2.3. The Lipid Composition of Membranes in HeLa–Kyoto Cells during Paclitaxel Chemotherapy

In order to verify the origin of the observed alterations in microviscosity, we performed an analysis of the lipid composition of cell membranes at different times of incubation with paclitaxel using ToF-SIMS.

Figure 4 reveals secondary ion yield variation of fatty acids fragment ions originating from membrane lipids for paclitaxel-exposed cells compared to a control sample. During incubation with paclitaxel, the proportion of unsaturated fatty acid chains increases by 15% after 1 h and 40% after 24 h incubation with the drug. The proportion of polyunsaturated fatty acids also increases by 11% after 1 h and 21% after 24 h, respectively. The recorded increase in the content of mono- and polyunsaturated fatty acids could explain the decrease in viscosity found at 1 and 24 h of incubation with the drug. In addition, an increase in the phosphatidylcholine signal of 44% was recorded in the first hour, which may also contribute to the decrease in the observed membrane microviscosity during chemotherapy with paclitaxel.

At the same time, the content of sphingomyelin remained unchanged, and the cholesterol signal increased after 24 h of incubation with paclitaxel. It is likely that this increase in cholesterol is a non-specific response of cells to drug stress since we observed such an increase in our previous studies with other chemotherapy drugs, for example, platinum-containing agents.

To identify differences in the mass spectra of different samples, principal component analysis (PCA) is often utilized, which was also used in this study. The results of the PCA showed only minor differences between the control and 24 h cells for both the positive and the negative ion polarities modes(Appendix A). The chemical profiles of control samples and samples of 1 h of incubation turned out to be quite similar.

The results of measurements of viscosity and lipid composition in cancer cells after paclitaxel treatment are summarized in Appendix A.

## 3. Discussion

It is known that the plasma membrane plays a critical role in the response of tumor cells to chemotherapy, but the relationships between membrane lipid composition, its biophysical properties, and drug response are poorly explored. In the present study, we have identified the effects of antimicrotubule agent paclitaxel on the plasma membrane of cancer cells in cell monolayer and tumor spheroids using a combination of microviscosity imaging by FLIM and membrane lipid profile analysis by ToF-SIMS.

The cytotoxic effect of many chemotherapy agents is, at least partly, associated with their contribution to the molecular organization of membranes, specifically with disrupting the organization of lipids. The plasma membrane is one of the most important targets, in addition to microtubules, for taxane-based anticancer drugs, including paclitaxel. These drugs can modify the physicochemical properties of membranes, such as fluidity (the reciprocal of viscosity) [13], the conformation of membrane-bound enzymes and receptors [14], lipid packing density [15], and lipid–lipid or lipid–protein interactions [16].

The relationship between membrane viscosity and the response of tumor cells to paclitaxel has been especially poorly studied. The viscosity of membranes plays an important role in the rate of diffusion and absorption of the drug into the cell. There are several studies pointing to the direct effect of paclitaxel on the viscosity of membranes through direct interaction with lipids. For example, using the methods of differential scanning calorimetry and electron paramagnetic resonance spectroscopy, a decrease in membrane viscosity was found in model membranes when paclitaxel was incorporated [17]. Zhao et al. also showed an increase in membrane fluidity with paclitaxel on model membranes [18].

Some other studies suggest the direct interaction of paclitaxel with membrane lipids, which can cause their liquefaction. For example, Kang and Loverde report on the interactions between hydrophobic paclitaxel and a model cell membrane at the molecular level using molecular dynamics simulations of all paclitaxel atoms. It has been shown that the main taxane ring is localized in the outer hydrophobic zone, while the three phenyl rings prefer to be located closer to the hydrophobic core of the membrane [19]. Süleymanogluit et al. found that paclitaxel liquefies the upper part of the acyl chains as it binds to the surface of phospholipid acyl chains and increases fluidity in the region of their head group, which also indicates a decrease in the viscosity of the lipid layer of cell membranes [20]. Moreover, paclitaxel has been shown to form temporary pores in the membrane, which also favor its fluidification. These facts can explain the liquefaction of tumor cell membranes when exposed to paclitaxel.

Another possible reason for the lowering of the microviscosity of membranes in paclitaxel-treated cells is alterations in the cellular cytoskeleton. The main mechanism of action of paclitaxel is associated with the ability of the drug to stimulate the assembly of microtubules from dimeric tubulin molecules and, therefore, to stabilize their structure, which disrupts the mitotic function of cells. There are many studies that show changes in microtubules at doses of paclitaxel close to those used in our study [21,22,23]. With regard to resistant cells, it was demonstrated that they had less polymerized tubulin and an increased growth rate of microtubules [24].

The association of tubulin with the plasma membrane involves several levels of penetration into the bilayer: from integral membrane proteins which bind on the surface to microtubules attached by linker proteins to proteins in the membrane [25]. In agreement with our study using paclitaxel, Aszalos et al. showed that depolymerization of microtubules using colcemid, colchicine, vinblastine, podophyllotoxin, and griseofulvin resulted in a decrease in the viscosity of the plasma membranes of CHO cells [26]. Rémy-Kristensen et al. investigated the effects of microtubule network integrity on plasma membrane fluidity in L929 mouse fibroblasts [27]. It was shown that in the cells treated with drugs that depolymerized microtubules, membrane fluidity increased, as assessed by fluorescence depolarization using a TMA-DPH probe. A slight decrease in microviscosity under the action of colchicine was also demonstrated on isolated membranes by Berlin et al. [28]. These studies suggest that microtubule integrity contributes to the high lipid order of the plasma membrane, but to a lesser extent than other factors such as lipid composition and cholesterol content.

It is known that paclitaxel affects not only tubulin but also the actin cytoskeleton [23]. There are data indicating the relationship between the submembrane actin cytoskeleton and the biophysical parameters of the membrane, including its microviscosity [27,29]. For example, the movements of integral membrane proteins are more limited than those of lipids, mainly due to interactions with the cytoskeleton through their intracellular domains [29]. So, they anchor in the membrane and serve as stabilizers of its fluid properties. On the other hand, the cytoskeleton can form barriers directly under the plasma membrane, and the organization of the cytoskeleton under the membrane limits the area of the membrane where molecules, including anticancer agents, can diffuse. An attempt to evaluate the cytosol viscosity under the action of paclitaxel and to correlate it with changes in the cytoskeleton structure was made by Chen et al. Using the method of optical tweezers, it was shown that the viscosity of the cytoplasm of cancer cells treated with paclitaxel dramatically decreases in the first 3 h due to the destruction of actin filaments [30]. In general, the effect of submembrane cytoskeleton organization on membrane microviscosity is not fully understood, but it can not be ruled out that alterations of actin structure also contributed to the fluidization of the membrane upon treatment with paclitaxel. Analysis of the cytoskeleton rearrangements in paclitaxel-treated cancer cells was beyond the scope of this study.

In our study, we noticed a trend towards a decrease in the microviscosity of tumor cell membranes in the first hours of incubation with paclitaxel and a dramatic drop in microviscosity after 24 h. In general, our results are in good agreement with the literature on membrane fluidification under the action of paclitaxel.

Fluidification of the plasma membrane is involved in the induction of apoptosis [31,32]. In our study, the recorded decrease in membrane viscosity is consistent with the presence of extensive areas of cell death, including late stages of apoptosis, detected by PI staining.

The viscous properties of cell membranes largely depend on the qualitative and quantitative composition of lipids in these membranes. Depending on the lipid profile, the effectiveness of paclitaxel may vary. Specifically, the addition of cholesterol decreases the penetration of paclitaxel due to a decrease in diffusion, which was demonstrated on the model membranes [6,33]. In a study by Pereira et al., it was also shown that the effects of paclitaxel on Langmuir monolayers strongly depend on the relative concentration of cholesterol in the membrane [33]. To evaluate the role of phospholipid properties, Zhao et al. examined the penetration profiles of paclitaxel in model membranes of different compositions [34]. Lipid chain length has been shown to affect drug–membrane interactions, and while shorter chain length phospholipids have shown a higher ability to incorporate paclitaxel, longer chains have hindered the penetration process due to higher packaging hence acting as a physical barrier. In addition, saturated phospholipids may pack tightly into each other, resulting in less incorporation of paclitaxel.

In our study, it was shown, for the first time, that paclitaxel treatment changes the composition of membrane lipids. Mass spectrometric analysis revealed a statistically significant increase in the signal of phosphatidylcholine in the first hour of exposure to paclitaxel. Xia Li and Ying-Jin Yuan showed, using the NPLC-ESI/MS(n) procedure, that changes in some phosphatidylcholine species with unsaturated fatty acid chains and phosphatidic acid species with saturated fatty acid chains are closely related to the change in cell membrane fluidity that occurs during apoptosis [35]. In addition, we noticed an increase in the content of mono- and polyunsaturated fatty acids after treatment with paclitaxel, which correlated with decreased microviscosity. Menendez et al. have shown that polyunsaturated fatty acids (PUFA) are able to enhance the sensitivity of cancer cells to paclitaxel by enhancing peroxidative processes and regulating the expression of oncoproteins [36]. The most potent PUFA that increased the toxicity of paclitaxel in breast cancer cell lines was gamma-linolenic acid. The most common event characterizing multidrug-resistant cells is the overexpression of ABC transporters such as P-glycoprotein (Pgp). In several studies, PUFAs are described as direct inhibitors of Pgp, and an increase in their concentration can suppress the activity of Pgp. For example, ω-3 and ω-6 PUFAs reduced Pgp transcription in colorectal cancer cells [37]. Thus, an increase in the content of unsaturated fatty acids may be a part of the response of the tumor cell to drug stress.

Previous studies have revealed that different drugs induce different alterations to cell membranes. For example, platinum drugs (cisplatin, oxaliplatin) initially caused a decrease (minutes to 1 h) and then an increase (1–2 days) in the microviscosity of plasma membranes, the latter mainly associated with an increase in cholesterol content [10,11] but not with a partitioning of the drug into the cell membrane. The drug 5-fluorouracil, a cytotoxic agent of the anti-metabolite class, induced only short-term fluctuations in membrane viscosity immediately after the addition of the drug, whereas the development of resistance to 5-fluorouracil resulted in a steady increase in viscosity associated with an increase in the content of sphingomyelin and cholesterol [12]. Unlike platinum drugs and 5-fluorouracil, paclitaxel has a different intracellular target, which causes completely different changes to membrane microviscosity.

The effects of different drugs on membrane viscosity and the lipid profile of cancer cells are summarized in Appendix A.

Based on our results with platinum drugs, 5-fluorouracil, and paclitaxel, we can suggest that the microviscosity of the cell membrane is regulated not by one type of lipid but by a complex interplay between different lipids that constitute the membrane. Notably, among the observed alterations in lipid composition induced by these drugs, there were always those that correlated with the viscosity changes.

The plasma membrane is a highly heterogeneous structure that includes the bulk lipid bilayer and “lipid rafts.” It is known that the bulk liquid-phase plasma membrane contains less cholesterol and sphingomyelin and more phospholipids with unsaturated acyl chains compared to the lipid rafts. The lipid rafts are the cholesterol-rich, highly ordered lipid “islands” that act as organizing hubs for membrane-embedded proteins. A limitation of our study is that the ToF-SIMS technique measures the signal from the entire cell surface, including lipid rafts, while microviscosity is measured only in the hydrophobic lipid bilayer of the membrane, where the BODIPY 2 rotor is localized. We suppose that this could be a reason for discrepancies between microviscosity and lipids (especially cholesterol and sphingomyelin) changes in the treated cells.

The precise mechanisms of membrane fluidization upon paclitaxel treatment in cellular models have yet to be clarified.

## 4. Materials and Methods

### 4.1. Cell Culture

HeLa–Kyoto (human cervical cancer) cells were used. The cells were cultured in DMEM (Life Technologies, Carlsbad, CA, USA) containing 100 μg/mL penicillin, 100 μg/mL streptomycin sulfate, and 10% fetal bovine serum (FBS) at 37 °C in a humidified atmosphere with 5% CO_2_.

To generate tumor spheroids, HeLa–Kyoto cells were seeded into ultra-low attachment 96-well round bottom plates (Corning Inc., Corning, NY, USA), ~300 cells/well in 200 μL DMEM and cultured in standard conditions (37 °C, 5% CO_2_, 80% humidity). After 5 days, spheroid formation was verified using light microscopy.

### 4.2. Chemotherapy

The chemotherapeutic drug Paclitaxel (Bristol–Myers Squibb, Irving, TX, USA) was used at doses of 1.6 nM (IC50/2), 3.2 nM (IC50), and 6.4 nM (2× IC50) for monolayer HeLa–Kyoto cells. The IC50 concentration was determined in our previous experiments using the MTT assay. Spheroids were treated with 6.5 nM of paclitaxel. The cells were incubated with the drug from 10 min to 48 h, spheroids from 3 to 48 h, and microviscosity was measured immediately after the treatment. Untreated cells or spheroids were used as a control.

The protocol for the creation of a paclitaxel-adapted cell subline was adopted from Ref. [38]. Briefly, a HeLa cell culture was continuously exposed to gradually increasing concentrations of paclitaxel. The initial concentration was 1/150 IC50. Each successive concentration was increased by 25% from the previous one and added only clear adaptation of the cells to the drug, i.e., after restarting cell proliferation without significant cell death in a plate (after 2–7 days). In ~4 months from first exposure, the cells were considered paclitaxel-resistant. Measurements of microviscosity in these cells were performed in 48 h after washing out the drug to avoid the immediate effects of paclitaxel.

### 4.3. Cell Viability Assay

A live/dead double staining kit (Sigma)—Calcein AM and propidium iodide (PI)—was used to stain live and dead HeLa–Kyoto cells, respectively, after chemotherapy according to the manufacturer’s protocol [39]. Cells were stained after 24 h, spheroids after 3, 6, 24, and 48 h, and the percentage of dead cells (stained with PI) of the total number of cells was calculated. The fluorescence of calcein was excited using an argon laser at a wavelength of 488 nm, and the emission was measured in the range of 500–570 nm. PI fluorescence was excited at a wavelength of 543 nm, and the emission was measured in the range of 600–700 nm. One-photon fluorescence confocal images were obtained using an LSM 880 (Carl Zeiss, Göttingen, Germany) laser scanning microscope.

### 4.4. Fluorescent Molecular Rotor BODIPY 2 and FLIM Microscopy

BODIPY 2 (4,4-difluoro-4-bora-3a,4a-diaza-s-indacene) was used as a viscosity-sensitive probe [8,9]. For microscopic imaging, the cells were seeded on glass-bottom dishes for confocal microscopy (FluoroDishes, Life Technologies, Carlsbad, CA, USA) in complete DMEM media without phenol red (Life Technologies, USA). The membrane viscosity was examined at 10 min, 1, 3, 6, and 24 h after adding Paclitaxel. Before imaging, the culture media was replaced with ice-cold Hank’s solution without Ca^2+^/Mg^2+^, and cells were incubated at +4 °C for 3 min. Thereafter, Hank’s solution was replaced with an ice-cold BODIPY 2 solution (4.5 μM). The spheroids were stained with BODIPY 2 on day 5 after the growth phase when they had a compact heterogeneous structure and a size of ~330 μM. A total of 4–6 spheroids were carefully transferred to each glass-bottom dish in 1.5 mL DMEM without phenol red and placed in a CO_2_ incubator for 1 h to allow their attachment. The membrane viscosity was examined at 3, 6, 24, and 48 h after adding paclitaxel. Before imaging, the culture media with paclitaxel was replaced with ice-cold Hank’s solution without Ca^2+^/Mg^2+^, and cells were incubated at +4 °C for 10 min. Afterward, Hank’s solution was replaced with an ice-cold BODIPY solution (8.9 μM). Two-photon excited FLIM images were acquired within 5–10 min after adding BODIPY2, while the probe was located in the plasma membranes.

A multiphoton tomography MPTflex (JenLab, Jena, Germany) with a tuneable 80 MHz, 200 fs Ti:Sapphire laser (MaiTai), a single-photon counting module SPC-150 and detector PMC-100-20 (Becker&Hickl, Berlin, Germany), as well as an LSM 880 (Carl Zeiss, Jena, Germany) laser scanning microscope equipped with a FLIM module, the SPC 150 TCSPC (Becker & Hickl GmbH, Berlin, Germany) and a Mai Tai HP femtosecond laser, 80 MHz, 140 fs (Spectra Physics, Milpitas, CA, USA) were used for fluorescence lifetime imaging microscopy (FLIM).

Using MPTflex, the images were acquired through a 40×, 1.3 NA oil immersion objective. In the case of spheroids, the images were acquired from a depth of ~20 μm. BODIPY 2 fluorescence was excited at a wavelength of 850 nm and detected in the range of 409–680 nm using a fixed pre-fitted emission filter. The average power applied to the sample was ~7 mW. The acquisition time was ~7 s per image. On the LSM 880 microscope, BODIPY 2 fluorescence was excited at a wavelength of 850 nm and detected in the range of 500 to 550 nm. A C Plan-Apochromat 40×/1.2 NA objective lens was used for image acquisition. The FLIM images were acquired at a laser power of 1–2%, with a photon collection time of 60 s.

Fluorescence lifetime analysis was performed in the SPCImage 8.3 software (Becker & Hickl, Germany). The collected amount of photons per the decay curve was at least 5000. FLIM images were obtained from 10 randomly selected fields of view in each culture dish. Fluorescence decays at each pixel of the whole image were fitted using a monoexponential model. The goodness of the fit χ^2^ ≤ 1.20 indicates that the model used provided a reasonable fit. The fluorescence lifetime of BODIPY 2 was measured in the plasma membranes of cells by manually selecting zones of the plasma membrane as regions of interest.

Experimentally measured lifetimes of BODIPY 2 (in ns) were converted to viscosity values (in cP) using previously obtained calibration plots [40].

### 4.5. ToF-SIMS

HeLa–Kyoto cells (5 × 10^5^) were seeded on confocal dishes containing clean and dry poly-L-lysine-coated cover glass and were incubated at 37 °C and in an atmosphere of 5% CO_2_ for 24–48 h. Then, paclitaxel (3.2 nM) was added to the culture medium. After 1 or 24 h incubation with paclitaxel, cells were washed three times with phosphate buffer (PBS), and then cells were incubated with 4% paraformaldehyde (PFA) for 45 min at room temperature for chemical fixation. Afterward, cells were washed three times with PBS. In total, three samples were prepared—control samples without drugs and cells incubated with paclitaxel for 1 h and 24 h. Fixed cells were stored in PBS for ~5–6 h due to shipping to the ToF-SIMS laboratory. Cells were washed with mQ water to remove excess salts. Drying was carried out under a gentle stream of argon at room temperature for 2 h.

Mass spectra were acquired with a ToF-SIMS 5 (ION-TOF Gmbh, Münster, Germany) equipped with a 30 keV Bi_3_^+^ liquid metal ion source. Twelve mass spectra were recorded for each sample in both positive and negative ion polarities, with an analysis area of 300 × 300 µm^2^, and the raster was 64 × 64 pixels. The primary ion dose density was 43 × 1011 ions/cm^2^ to maintain static SIMS conditions. A low-energy electron flood gun was activated for charge compensation in all experiments. Lipid ion yields were calculated from the intensity of the corresponding peak of interest normalized to the total ion count amount.

The areas within a sample with morphological and compositional anomalies, such as dead cells and contamination particles, were excluded from the analysis. Moreover, dead cells, if present in a sample, are detached during sample preparation for ToF-SIMS. So the lipids analysis was performed mostly on a viable cell population.

### 4.6. Statistics

The mean values (M) and standard deviations (SD) were calculated for the microviscosity values. Student’s *t*-test was used to compare data (*p* < 0.05 was considered statistically significant). The number of cells for mean value calculations was 50–70 in 7–10 fields of view.

## 5. Conclusions

Integrity and fluidity of biological membranes are critical for cellular homeostasis. The evolution of the membrane–lipid therapy of malignant tumors, a novel promising approach to anticancer treatment, stimulates investigations of the effects of conventional cytotoxic drugs on biophysical properties and molecular composition of cell membranes. Recent studies suggest that chemotherapeutic agents are able to alter membrane fluidity, and the drugs of different classes affect the membrane state in different ways, even if they have a common intracellular target (e.g., DNA). Our results obtained on cancer cell monolayers and tumor spheroids indicate, for the first time, that chemotherapy with paclitaxel induces membrane lipid remodeling and associated alterations of microviscosity. The most pronounced effect was a fluidization of the membrane at prolonged (>24 h) exposures, which occurred, most likely, due to the high level of unsaturated fatty acids. Since fluidity determines the passive permeability of the membranes, its changes due to drug treatment can have an effect on the cellular accumulation of other drugs used in combination and, therefore, should be taken into account in the planning of therapeutic regimens. In addition, further pharmacological decreases in plasma membrane viscosity may sensitize cancer cells to paclitaxel and, thus, improve treatment outcomes.

## Figures and Tables

**Figure 1 ijms-24-12186-f001:**
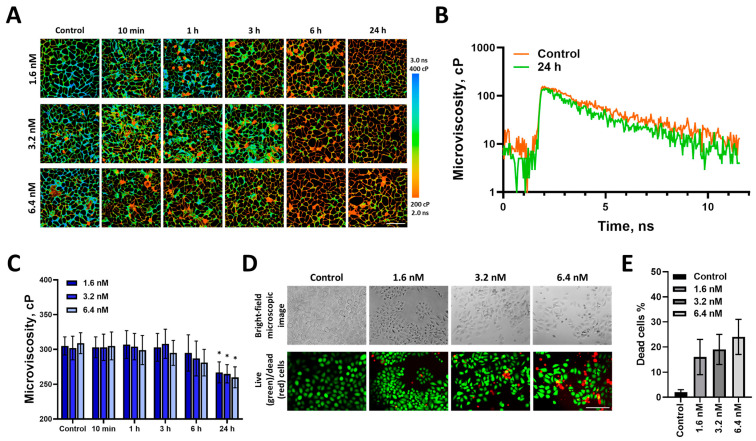
Microviscosity of plasma membrane in monolayer HeLa cells before (control) and during 24 h exposure to different doses of paclitaxel. (**A**) FLIM images of cells stained with BODIPY2. Bar, 40 μm. Viscosity measurements were performed only in live cells since non-monoexponential decay of the rotor was observed in dead cells (presumably due to its aggregation). (**B**) Representative decay curves of BODIPY 2 in plasma membranes of control and treated cells after 24 h exposure to paclitaxel (3.2 nM). (**C**) Quantification of microviscosity during chemotherapy. Mean ± SD, *n* = 60 cells. * *p* = 0.00011 compared with control. (**D**) Morphology and live (green)/dead (red) cells assay after 24 h incubation with paclitaxel. Bar, 80 μm. (**E**) Quantitative analysis of dead cells in control and treated cell populations, %.

**Figure 2 ijms-24-12186-f002:**
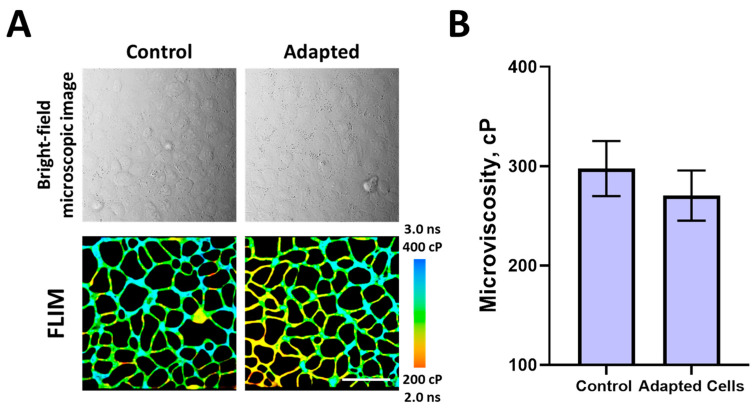
Membrane microviscosity in paclitaxel-adapted cells. (**A**) Bright-field images and FLIM images of cells. Bar, 40 μm. (**B**) Quantification of microviscosity in control and adapted cells. Mean ± SD, *n* = 80 cells.

**Figure 3 ijms-24-12186-f003:**
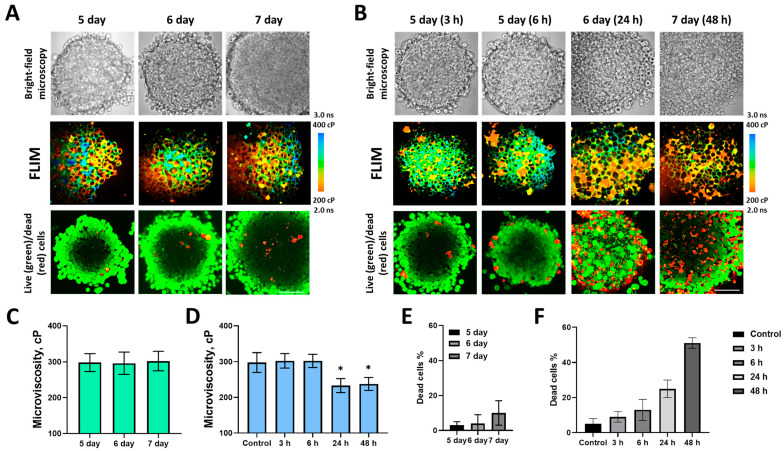
Plasma membrane microviscosity in HeLa tumor spheroids before (control) and during 48 h exposure to paclitaxel. Bright-field, FLIM images and live (calcein)/dead (PI) cells assay of control (**A**) and treated (**B**) spheroids. Bar, 80 μm. Quantification of microviscosity during spheroid growth in control (**C**) and treated (**D**) groups. Mean ± SD, *n* = 4 spheroids, 60 cells in each. * *p* < 0.05. Quantitative analysis of dead cells in control (**E**) and treated (**F**) spheroids, %.

**Figure 4 ijms-24-12186-f004:**
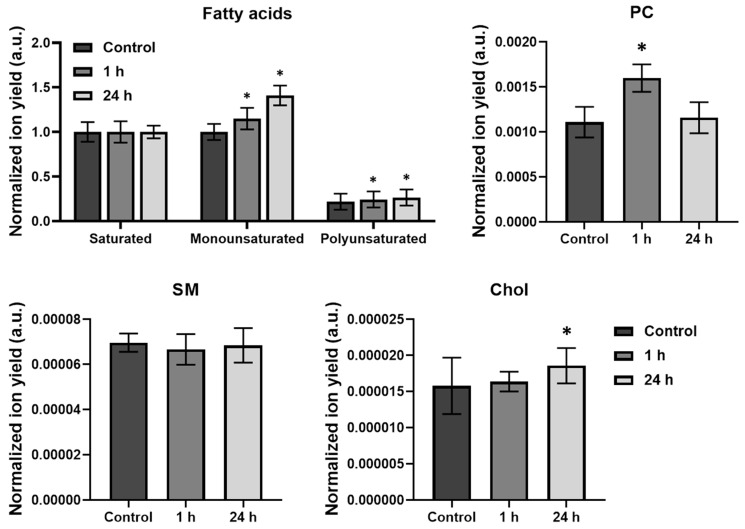
Analysis of lipid composition of HeLa cell’s membranes during paclitaxel treatment. Comparison of secondary ion yields of fatty acids, phosphatidylcholine (PC) (*m/z* 224), sphingomyelin (SM) (*m/z* 264), and cholesterol (Chol) (*m/z* 385) yield after 1 h and 24 h incubation with paclitaxel. * Statistically significant difference with control, *p* < 0.01.

## Data Availability

The data that support the findings of this study are available from the corresponding author upon reasonable request.

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
