# Peer review of "Effects of Paclitaxel on Plasma Membrane Microviscosity and Lipid Composition in Cancer Cells"

_ijms, 2023, doi:10.3390/ijms241512186_

Round 1

Reviewer 1 Report

Recommendation:

Minor review

Comments:

  The approach taken in this paper is novel in that it combines spectroscopy and mass spectrometry to analyze changes in cell membranes in response to chemotherapeutic drug treatment. The study of the biochemical and biophysical effects occurring in the cell membrane upon treatment with paclitaxel, revealing changes in cell membrane components and microviscosity. Especially the use of fluorescence lifetime imaging microscopy (FLIM) and the fluorescent molecular rotor BODIPY 2 to visualize the microviscosity of the cell membrane was interesting. In addition, the lipid profile of the membrane, assessed using the time-of-flight secondary ion mass spectrometry (ToF-SIMS), seems appropriate as a chemical approach. The results presented not only the changes in the microviscosity of the membranes in both cell monolayers and tumor spheroids after treatment, but also the changes in the types of fatty acids were impressive. The paper would be more complete if some minor improvements were made.

Additional comments:

1.      Page 2, line 56 typo ; meaningless bracket

2.      In the introduction, it is mentioned that fluorescence lifetimes can be directly converted to the viscosity of the medium using a pre-recored calibration curve, but it would be nice to specify what the lifetimes of the fluorescence in Fig. 1A are color-matched to and how these changes translate to the viscosity of the medium (Fig. 1C) so that the reader can understand them directly without inference.

3.      I am recommending that the group-specific information for the ToF-SIMS lipid candidates summarized and described in figure 4 be tabled and added to the supplyment information. 

4.    Changes in cell membrane viscosity in response to drug treatment can be explained, but could the altered lipids identified by SIMS in the spheroid model be from dead cells and not from the drug effect? 

Date of report: July. 14, 2023

Reviewer 3 Report

In the present study, changes in the microviscosity of membranes of living Hela Kyoto tumor cells were studied during chemotherapy with paclitaxel, a widely used antimicrotubule agent.
1. Some keywords are long.
2. In line 56, add the reference to Mer Bouta.
3. Lines 33 to 42 do not have references.
4. The manuscript should be updated with new references (2021-2023).
5. No reference is given for the method of Cell viability assay.

6. Why were the different doses of Paclitaxel (1.6, 3.2, 6.4 nM) chosen?

7. Comparison of this study with other studies in the discussion is not done.

Minor editing of English language required

Round 2

Reviewer 3 Report

All corrections have been made and the manuscript is acceptable.